energy/chemical engineering

diffusion coefficient, supercritical CO$_2$, porous media, reservoir condition, empirical correlation

**Author for correspondence:**
Yi Zhang
e-mail: zhangyi80@dlut.edu.cn

# Experimental study of the supercritical CO$_2$ diffusion coefficient in porous media under reservoir conditions

Junchen Lv[†], Yuan Chi[†], Changzhong Zhao, Yi Zhang and Hailin Mu

Key Laboratory of Ocean Energy Utilization and Energy Conservation Ministry of Education, School of Energy and Power Engineering, Dalian University of Technology, Dalian 116024, People's Republic of China

(iD) JL, 0000-0003-0057-0874; YC, 0000-0003-1530-1503

Reliable measurement of the CO$_2$ diffusion coefficient in consolidated oil-saturated porous media is critical for the design and performance of CO$_2$-enhanced oil recovery (EOR) and carbon capture and storage (CCS) projects. A thorough experimental investigation of the supercritical CO$_2$ diffusion in $n$-decane-saturated Berea cores with permeabilities of 50 and 100 mD was conducted in this study at elevated pressure (10–25 MPa) and temperature (333.15–373.15 K), which simulated actual reservoir conditions. The supercritical CO$_2$ diffusion coefficients in the Berea cores were calculated by a model appropriate for diffusion in porous media based on Fick's Law. The results show that the supercritical CO$_2$ diffusion coefficient increases as the pressure, temperature and permeability increase. The supercritical CO$_2$ diffusion coefficient first increases slowly at 10 MPa and then grows significantly with increasing pressure. The impact of the pressure decreases at elevated temperature. The effect of permeability remains steady despite the temperature change during the experiments. The effect of gas state and porous media on the supercritical CO$_2$ diffusion coefficient was further discussed by comparing the results of this study with previous study. Based on the experimental results, an empirical correlation for supercritical CO$_2$ diffusion coefficient in $n$-decane-saturated porous media was developed. The experimental results contribute to the study of supercritical CO$_2$ diffusion in compact porous media.

## 1. Introduction

Global warming caused by the excessive emission of greenhouse gases (mainly CO$_2$) has recently attracted much

[†]Co-first authors.

attention. Many techniques have been proposed to mitigate $CO_2$ emissions [1–3]. Carbon capture and storage (CCS) is one of the most promising options to reduce the atmospheric $CO_2$ concentration [4–6]. Injection of $CO_2$ into oil reservoirs known as $CO_2$-enhanced oil recovery (EOR) can not only help to store the $CO_2$ but also improve the recovery of crude oil [7,8]. Hence, this process is considered to be the most cost-effective CCS technique and has been widely used in the development of oilfields [9–11].

During the $CO_2$-EOR process, the diffusion of $CO_2$ into crude oil results in volumetric expansion and viscosity reduction of oil in the reservoir [12–14]. Therefore, the determination of $CO_2$ diffusion coefficient in the porous media saturated with oil has extremely important guidance on $CO_2$-EOR risk assessment, engineering design and economic evaluation [15–17].

Many studies have been conducted on the diffusion coefficient measurement using the pressure volume temperature (PVT) method since the 1930s [18–22]. The PVT method is usually coupled with pressure decay method to determine the diffusion coefficient. A PVT cell with constant volume is used as diffusion cell. The pressure of the system is measured and recorded in real time. A pressure–time profile is then obtained and applied with different mathematical models to predict the diffusion coefficient. Renner proposed measuring the diffusion coefficient of $CO_2$ and ethane in consolidated porous media using a novel *in situ* method [15]. The experimental diffusion coefficients of $CO_2$ and ethane in decane were determined for conditions of 5.86 and 4.14 MPa at 311.15 K, respectively. Riazi used a constant volume diffusion cell to measure the diffusion coefficients of the methane-$n$-pentane system at a constant temperature of 310.95 K [18]. The pressure–time profile was obtained as the $CH_4$ diffused into the liquid phase. The pressure decay method was applied to measure the diffusion coefficient. Unatrakarn *et al.* used a similar technique to measure the $CH_4$ and $CO_2$ diffusion coefficients in porous media and bulk oil phase at pressures up to 3.2 MPa [23]. Li and Dong proposed both experimental and mathematical methods to determine the $CO_2$ diffusion coefficient in oil-saturated Berea cores [24]. The $CO_2$ diffusion coefficients were measured under a pressure range from 2.3 to 6.5 MPa. Li *et al.* used a similar method to measure the diffusion coefficient of $CO_2$ in cores saturated with oil obtained from the Shengli Oilfield in their experiments [25]. The temperature and permeability conditions were approximately 403.15 K and 9 mD, respectively. The experimental pressures in previous studies were usually less than 6 MPa, which is not consistent with the actual formation conditions.

In the present study, a comprehensive experimental investigation was conducted to investigate the effects of temperature, pressure, permeability, $CO_2$ state and porous media on the supercritical $CO_2$ diffusion coefficient in porous media saturated with $n$-decane. Experimentally, the pressure data for supercritical $CO_2$ diffusion process in $n$-decane-saturated Berea core of 23 experiments at elevated temperatures and pressures were measured. The ranges of temperature and pressure were from 10 to 25 MPa and 333.15 to 373.15 K, respectively. Theoretically, a simplified mathematical model of Fick's Law for the diffusion process in porous media was employed to determine the diffusion coefficients of all 23 experiments. Moreover, a pressure–temperature–viscosity-based empirical equation for the supercritical $CO_2$ diffusion coefficient in hydrocarbon-saturated porous media was developed. The main objective of this work was to contribute to the experimental study of supercritical $CO_2$ transport in consolidated porous media under reservoir conditions by providing exhaustive experimental data.

# 2. Experimental section

## 2.1. Materials

Berea cores with different permeabilities of 50 and 100 mD were used to examine the effect of permeability on the diffusion coefficient. The porosities of the 50 mD core and 100 mD core were 16% and 22.5%, respectively. The top and bottom faces of the core were sealed by epoxy resin to ensure that the diffusion process only occurred from the radial direction through the side face. Pure $CO_2$ gas (99.999%) was purchased from Dalian Special Gas Co. Ltd, China. Pure $n$-decane (98%) was used to represent the oil phase and was purchased from Macklin Biochemical Co., Ltd, China.

## 2.2. Experimental set-up

Figure 1 represents the schematic diagram of the experimental set-up used in this study. An oil bath (CORID CD series, JULABO Inc., Germany) with an accuracy of $\pm 0.03$ K was used to control and

**Figure 1.** Schematic of the supercritical $CO_2$ diffusion experimental set-up.

maintain the temperature of the diffusion cell at the desired value. The pump (D250 L, Jiangsu Haian Oilfield Scientific Instrument Co., Ltd) is applied to syringe highly pressurized supercritical $CO_2$ into diffusion cell via intermediate containers and control the pressure of the entire PVT system. The temperatures of the diffusion cell were measured by a temperature transducer with an accuracy of $\pm 0.2$ K (JM618I, Jinming Instrument Co., Ltd, China). The pressures were measured using a transducer with an accuracy of $\pm 0.02$ MPa (UNIK 5000, GE Druck Ltd, Germany). The change of pressure in the diffusion cell was recorded by a computer in real time.

## 2.3. Experimental process

The measurements of the supercritical $CO_2$ diffusion in $n$-decane-saturated Berea cores were performed with the experimental set-up shown in figure 1. The experimental steps were as follows:

(1) The Berea core was fully immersed in a beaker full of $n$-decane and was evacuated for 24 h with a vacuum pump to ensure that the core was fully saturated by $n$-decane.
(2) After the core was positioned in the diffusion cell, the diffusion cell was evacuated for 15 min by a vacuum pump to ensure a vacuum state was reached inside.
(3) The diffusion cell was then heated in the oil bath until the cell temperature reached the desired value.
(4) The $CO_2$ in the intermediate container was pressurized to a value 50% higher than the required value for experiments to ensure that the pressure inside the diffusion cell could reach the desired value quickly.
(5) The supercritical $CO_2$ was charged into the diffusion cell until the desired pressure was reached. The pressures of the diffusion process were measured by the pressure transducer and recorded by pre-installed software in real time.
(6) After the pressure of the diffusion cell reached steady state, the diffusion process was stopped. The remaining $CO_2$ was discharged from the diffusion cell. Then, both the core and diffusion cell were washed and dried to prepare for the next experiment.

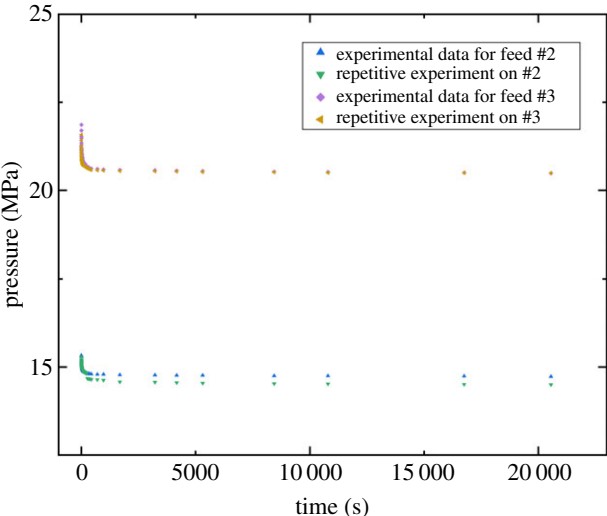

**Figure 2.** Repetitive experimental data.

## 3. Mathematical formulation

### 3.1. Assumptions

The following are the assumptions proposed for the mathematical model.

(1) In the experiments, the Berea cores were isotropic and $n$-decane was distributed uniformly in it.
(2) The effect of $n$-decane swelling was neglected in the experiments.
(3) During the diffusion process, the supercritical $CO_2$ diffusion coefficients were constant in the Berea cores.

### 3.2. Mathematical model

Based on the treatments to the cores mentioned previously, the diffusion process in this study became a natural convection mass transfer process induced by the diffusion of $CO_2$ into a liquid-saturated vertical porous column. The effective $CO_2$ diffusion coefficient under non-swelling conditions can be obtained from Fick's first Law and the continuity equation, as follows [25]:

$$\left.\begin{array}{c} \dfrac{\partial C}{\partial t} = D'_{eff}\left(\dfrac{\partial^2 C}{\partial r^2} + \dfrac{1}{r}\dfrac{\partial C}{\partial t}\right), 0 < r < r_0, t \geq 0, \\[2mm] C|_{t=0} = 0, 0 < r < r_0 \\[2mm] \text{and} \quad C|_{r=r_0} = C_0, t \geq 0, \end{array}\right\} \tag{3.1}$$

where $C$ denotes $CO_2$ concentration in the porous media, mol m$^{-3}$; $r$ denotes $CO_2$ diffusion radius, $0 < r < r_0$, m; $r_0$ denotes the core radius, m; $t$ denotes $CO_2$ diffusion time, $t \geq 0$, s; and $D'_{eff}$ denotes $CO_2$ diffusion coefficient, m$^2$ s$^{-1}$.

A simplified expression of the $CO_2$ diffusion coefficient under non-swelling conditions is shown in equation (3.2), which is obtained from other studies [26,27].

$$D'_{eff} = \frac{\pi}{16}\left(\frac{r_0 kV}{N_\infty ZRT}\right)^2, \tag{3.2}$$

where $k$ denotes the slope of straight line in the coordinate of pressure drop versus square root of time, as shown in figure 3$b$; $V$ denotes the annular volume between the core sample denotes gas constant, 8.314 J mol$^{-1}$ K$^{-1}$; $N_\infty$ denotes $CO_2$ mass diffused into the core when the diffusion time tends to infinity, mol.

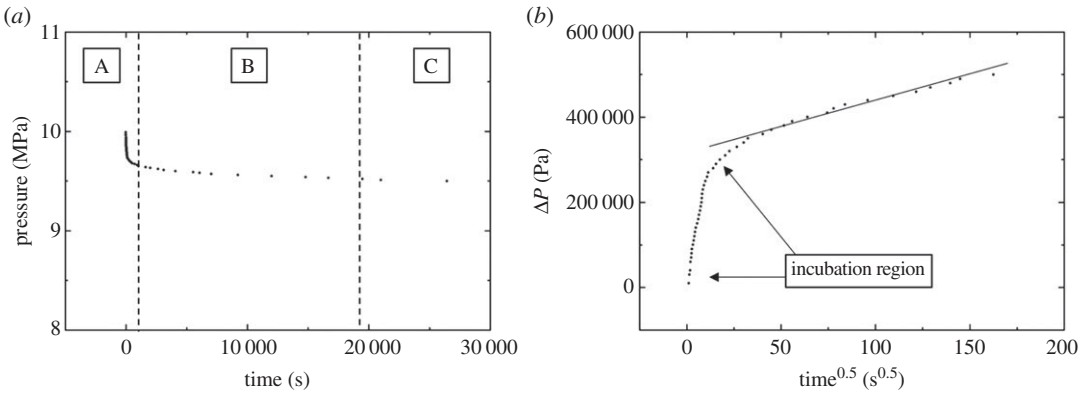

**Figure 3.** (*a,b*) Experimental data for Experiment No. 1

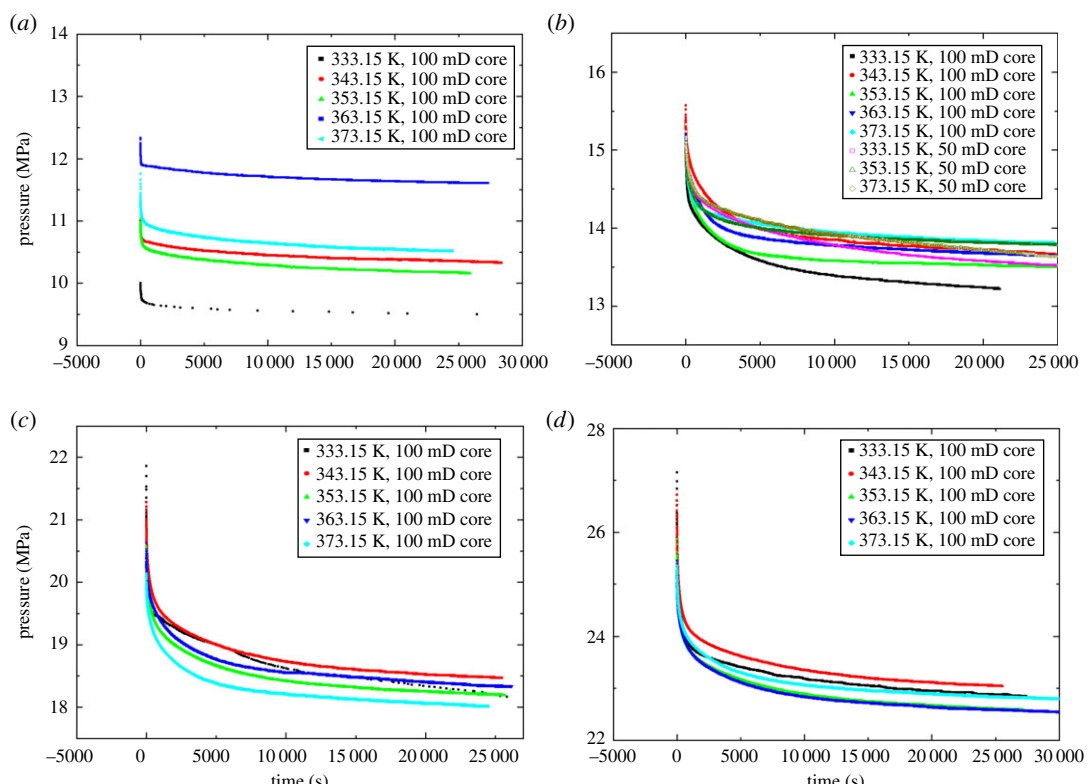

**Figure 4.** Pressure data from diffusion experiments with different initial pressures: (*a*) 10, (*b*) 15, (*c*) 20 and (*d*) 25 MPa.

## 4. Results and discussion

### 4.1. Validation of the experimental procedure

Figure 2 presents the results of the reproducibility test for the supercritical $CO_2$ diffusion process under pressures from 10 to 20 MPa and temperature at 333.15 K. Figure 2 shows that the original experiments were in good agreement with repetitive experiments. Similar tests were also used in other study to validate the reliability of the experimental apparatus [28].

The pressure data recorded during the supercritical $CO_2$ diffusion process for Experiment No. 1 are shown in figure 3. Figure 3*a* shows that the pressure–time (P–T) profile can be separated into three regions A, B and C corresponding to the rapid pressure decay region, the transition region and the steady pressure region, respectively. The steady state is indicated by the straight line following the inflection point in figure 3*b* which is a plot of the pressure drop as a function of the square root of time. Figure 4 presents the experimental pressure data for Experiments 1–23. Figure 4*b* includes the

**Table 1.** Summary of $CO_2$ diffusion coefficients and corresponding experimental conditions.

| experiments # | pressure (MPa) | temperature (K) | permeability (mD) | decane viscosity (cp) | diffusion coefficients ($10^{-10}$ m$^2$ s$^{-1}$) |
|---|---|---|---|---|---|
| 1 | 10.00 | 333.44 | 100 | 609.51 | 0.64 |
| 2 | 15.29 | 333.24 | 100 | 645.67 | 2.01 |
| 3 | 21.86 | 333.29 | 100 | 691.28 | 6.30 |
| 4 | 27.15 | 333.61 | 100 | 728.55 | 8.25 |
| 5 | 11.00 | 343.53 | 100 | 553.31 | 0.66 |
| 6 | 15.58 | 343.40 | 100 | 581.61 | 2.23 |
| 7 | 21.37 | 343.29 | 100 | 617.70 | 6.52 |
| 8 | 26.72 | 343.35 | 100 | 651.42 | 8.95 |
| 9 | 11.05 | 353.24 | 100 | 500.36 | 0.94 |
| 10 | 15.04 | 353.20 | 100 | 522.81 | 3.39 |
| 11 | 20.58 | 353.22 | 100 | 554.05 | 7.19 |
| 12 | 25.87 | 353.47 | 100 | 584.17 | 9.78 |
| 13 | 12.33 | 362.89 | 100 | 461.48 | 1.20 |
| 14 | 15.21 | 363.15 | 100 | 476.3 | 5.72 |
| 15 | 21.37 | 363.06 | 100 | 508.13 | 9.54 |
| 16 | 25.45 | 363.23 | 100 | 529.35 | 11.68 |
| 17 | 11.89 | 373.12 | 100 | 419.72 | 1.97 |
| 18 | 14.99 | 372.86 | 100 | 434.43 | 7.97 |
| 19 | 20.14 | 373.10 | 100 | 458.98 | 11.13 |
| 20 | 25.35 | 373.45 | 100 | 483.84 | 13.24 |
| 21 | 15.43 | 333.30 | 50 | 646.65 | 0.88 |
| 22 | 15.03 | 353.31 | 50 | 522.69 | 1.59 |
| 23 | 15.11 | 373.04 | 50 | 435.02 | 3.16 |

three diffusion experiments using the 50 mD Berea cores. The remaining experiments used cores with a permeability of 100 mD.

Single factor design was employed to study the effect of temperature, pressure and permeability on the supercritical $CO_2$ diffusion coefficient in *n*-decane. In each experiment, only one experimental condition was changed while the other two conditions were kept constant. All the experimental results are listed in table 1.

## 4.2. Empirical correlation for diffusion coefficient

According to the experimental results above, a pressure–temperature–viscosity-based empirical correlation was developed for the supercritical $CO_2$ diffusion coefficient in porous media saturated with *n*-decane. The correlation for Berea cores with a permeability of 100 mD is shown in equation (4.1). The R-square of this correlation is 0.9253. Figure 5 shows the comparison between the experimental results and the empirical correlation predictions.

$$D = 10^{-10} \times P^{3.1078} \times T^{0.9337} \times \mu^{-2.0558}, \tag{4.1}$$

where $D$ denotes the diffusion coefficient; $P$ denotes the pressure, MPa; $T$ presents the temperature, K; $\mu$ is hydrocarbon viscosity, cp.

## 4.3. The effect of pressure

Figure 6 shows that the diffusion coefficient of the supercritical $CO_2$ increases with pressure. The diffusion coefficient at 25 MPa is 12.99 times larger than the one at 10 MPa in 333.15 K experiments.

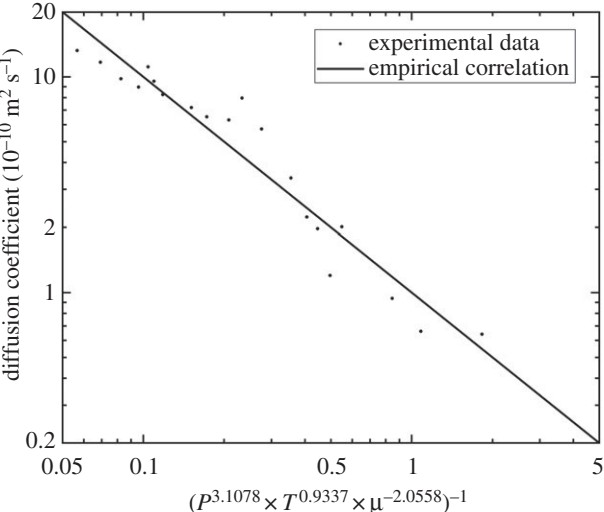

**Figure 5.** Comparison of experimental results with empirical correlation prediction.

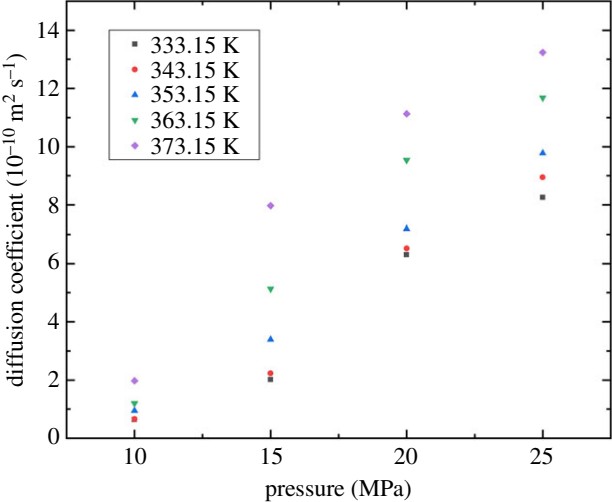

**Figure 6.** Effect of pressure on the $CO_2$ diffusion coefficient.

For the remaining four experimental temperatures from 343.15 K to 373.15 K, the diffusion coefficients increase by 13.47, 10.38, 9.72 and 6.70 times, respectively. This shows that the diffusion process will happen faster when the pressure is larger. The concentration of supercritical $CO_2$ in the diffusion cell is larger under higher pressure conditions. Thus, the diffusion coefficients increase with the supercritical $CO_2$ concentration due to a reduced viscosity of the supercritical $CO_2$–decane mixture. Similar experimental observations could be found in other studies [29,30].

## 4.4. The effect of temperature

Figure 7 shows the temperature effect on the supercritical $CO_2$ diffusion coefficient. The relationship between the diffusion coefficient and temperature is a parabola that is concave up. The thermal molecular motion is the dominant factor that affected the diffusion process. With increasing temperature, the molecular motion becomes more active so that the $CO_2$ diffusion process is enhanced.

## 4.5. The effect of permeability

Figure 8 shows that the diffusion coefficient increases with permeability. The measurements were performed under 15 MPa for all six experiments shown in figure 8. The diffusion coefficient increases by 2.28, 2.13 and 2.52 times at 333.15, 353.15 and 373.15 K, respectively. The permeability of the core

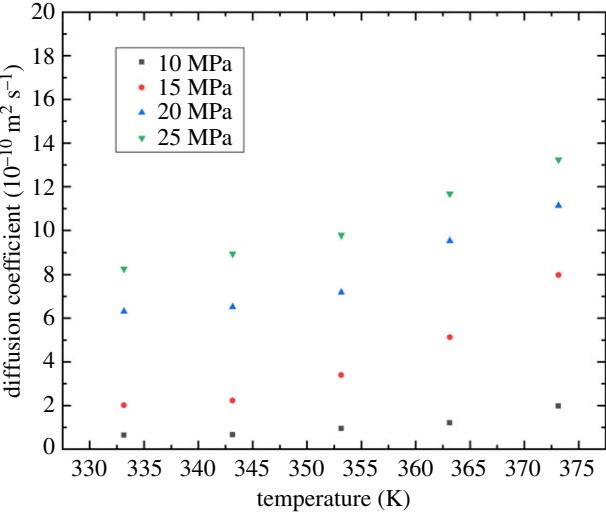

**Figure 7.** Effect of temperature on the $CO_2$ diffusion coefficient.

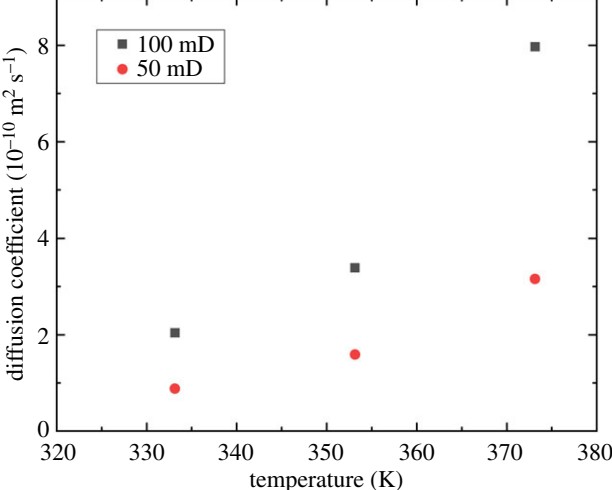

**Figure 8.** Effect of permeability on the $CO_2$ diffusion coefficient.

has a reciprocal relationship with its tortuosity. As a reflection on the diffusion coefficient, the increase in the permeability indicates a decrease in the tortuosity, which allows the supercritical $CO_2$ to flow more smoothly in the cores. As a result, the diffusion coefficient increases with the permeability.

## 4.6. $CO_2$ state and porous media effect

The experimental results of the $CO_2$ diffusion coefficient in different mixture systems and experimental conditions are listed in table 2. It is noted that the $CO_2$ diffusion coefficients from the previous study are generally larger than the results in this study. The differences in results were caused by the state of $CO_2$ in the different experiments. The $CO_2$ was in the vapour phase in both Grogan *et al.*'s and Bagalkot & Hamouda's studies [31,32]. In our study, $CO_2$ was in supercritical state with a larger viscosity and density. The diffusion process was inevitably restrained, which led to a decrease in the diffusion coefficient.

In Moultos *et al.*'s study, the $CO_2$ diffusion coefficients in bulk decane were obtained under similar experimental conditions [33]. The diffusion coefficient was $40 \times 10^{-10}$ m$^2$ s$^{-1}$ under 323.15 K and 30 MPa condition. The $CO_2$ diffusion coefficient in porous media saturated with decane obtained in this study was $8.25 \times 10^{-10}$ m$^2$ s$^{-1}$ under 333.61 K and 27.15 MPa condition. Under similar

**Table 2.** $CO_2$ diffusion coefficient under different conditions.

| mixture | experiment condition | diffusion coefficient ($10^{-10}$ m$^2$ s$^{-1}$) | reference |
|---|---|---|---|
| $CO_2$ + n-decane (porous media) | 311 K, 1.44 – 5.83 MPa | 104 – 126 | [15] |
| $CO_2$ + bulk n-decane | 298.15 – 318.15 K, 2.5 – 6 MPa | 12.1 – 22.6 | [31] |
| $CO_2$ + bulk n-heptane | 298.15 – 318.15 K, 2.5 – 6 MPa | 12.9 – 26.9 | |
| $CO_2$ + bulk n-hexane | 298.15 – 318.15 K, 2.5 – 6 MPa | 17.4 – 34.3 | |
| $CO_2$ + bulk pentane | 298.15 K, 1.54 – 3.51 MPa | 37.2 – 75.9 | [32] |
| $CO_2$ + bulk decane | 298.15 K, 1.36 – 5.63 MPa | 18.7 – 57.1 | |
| $CO_2$ + bulk hexadecane | 298.15 K, 2.26 – 5.28 MPa | 18.0 – 31.7 | |
| $CO_2$ + bulk n-decane | 298.15 – 323.15 K, 1 – 30 MPa | 25 – 48 | [33] |
| $CO_2$ + bulk octane | 290 – 311 K, 1.265 – 3.103 MPa | 2.789 – 8.105 | [34] |
| $CO_2$ + bulk n-tetradecane | 290 – 311 K, 0.910 – 4.041 MPa | 0.767 – 3.731 | |
| $CO_2$ + n-decane (porous media) | 333.15 – 373.15 K, 10 – 25 MPa | 0.64 – 13.24 | this study |

temperature and pressure conditions, the diffusion coefficient in bulk decane is 4.84 times larger than that in porous media. The result clearly shows that the $CO_2$ diffusion process is impeded significantly by the presence of porous media. The experimental results in this paper have better practical meaning since the pressure and temperature conditions simulated the underground reservoir conditions.

# 5. Conclusion

In this study, a comprehensive experimental investigation on the effects of the pressure, temperature, permeability, $CO_2$ state and porous media on the supercritical $CO_2$ diffusion coefficients was carried out. Overall, 23 experiments were performed at pressure ranging from 10 to 25 MPa and temperature ranging from 333.15 to 373.15 K, which simulated the real reservoir conditions. Among them, Experiments 1–20 used Berea cores with a permeability of 100 mD and Experiments 21–23 used cores with a permeability of 50 mD to study the effect of the permeability on the diffusion coefficient.

These results demonstrated that under reservoir conditions the supercritical $CO_2$ diffusion coefficient in oil-saturated porous media increased with increasing pressure and temperature, respectively. However, the effect of pressure on the diffusion process decreased at elevated temperature condition. Also, the supercritical $CO_2$ diffusion coefficient first increases slowly at 10 MPa and then grows significantly with increasing pressure. The increase in the core permeability increased the diffusion coefficient and the growth trend was steady throughout the whole temperature range in the experiments. The supercritical state of the $CO_2$ and the presence of porous media significantly impeded the diffusion process, compared with the results from pure gas $CO_2$ diffusion in bulk alkanes in previous study. The pressure–temperature–viscosity-based empirical equation developed in this paper successfully predicted the supercritical $CO_2$ diffusion coefficients in porous media under reservoir conditions. Moreover, all the experimental data contributed to the theoretical study of supercritical $CO_2$ diffusion in porous media at elevated temperatures and pressures.

Data accessibility. The measured experimental data are available from the Dryad Digital Repository: https://doi.org/10.5061/dryad.n0bf69r [35].

Authors' contributions. J.L. carried out the experimental measurements in the laboratory, processed the experimental data, participated in the design of the study and drafted the manuscript. Y.C. helped with the experimental measurement work in the laboratory. C.Z. helped with processing the experimental data. Prof. Y.Z. directed this study and helped drafted the manuscript. Prof. H.M. offered theoretical support and helped draft the manuscript. All authors gave final approval for publication.

Competing interests. We have no competing interests.

Funding. This paper was supported by the National Key R&D Program of China (grant no. 2016YFB0600804), National Natural Science Foundation of China (51576031 and 51436003) and the Fundamental Research Funds for the Central Universities (grant no. DUT18LAB22).

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
