## [Reviewer comments · Royal Society Open Science]

Review History

RSOS-181902.R0 (Original submission)

Review form: Reviewer 1

Is the manuscript scientifically sound in its present form?

Yes

Are the interpretations and conclusions justified by the results?

Yes

Is the language acceptable?

Yes

Is it clear how to access all supporting data?

Yes

Do you have any ethical concerns with this paper?

No

Have you any concerns about statistical analyses in this paper?

No

Recommendation?

Accept with minor revision (please list in comments)

Comments to the Author(s)

This paper presents experimental results measuring diffusion coefficients for supercritical CO₂ under a variety of pressure, temperature, and permeability conditions of Berea sandstone. These results add important data to other studies of supercritical CO₂ diffusion coefficients because the experiments were carried out under conditions relevant for CO₂ sequestration. The authors compile their results with very useful empirical relationship between the diffusion coefficient and pressure, temperature, and hydrocarbon viscosity. Such a laboratory-based relationship could prove very useful for numerical modeling studies of CO₂ sequestration, and therefore I recommend that these results be published.

The paper is written in a very straight-forward manner; "this is what we did, and this is what we measured." However, there are numerous grammatical and style problems, which could be fixed in revision. I list a few of them here:

line 42 (abstract): "...then grow...". Change to " then grows significantly with increasing pressure":

page 10.

line 4: change to "has recently attracted much attention".

line 23: Change "researches" to "studies". (apply this throughout the paper).

line 31: need citation after "Renner"

line 35: change "were 5.86...." to "were determined for conditions of 5.86..."

page 11.

line 14. Do not start sentence with And

page 13.

line 47: change to "is shown from other studies...".

page 14.

line 8: Change to "Figure 2 shows that the original...."

line 9: Change diffuse to diffusion

line 40: Change to "Figure 3a shows...."

page 15.

line 57. Feed is not capitalized (and not even sure if feed is the right word).

page 17.

line 43: change to "Figure 6 shows..."

line 57: change "literatures" to "studies"

page 19.

line 43: "different mixtures systems.."
line 44: change "were" to "are"
line 47-48. This sentence makes no sense
line 54: Moutos needs a citation

page 20.
line 5: change "had" to "have"

Review form: Reviewer 2

Is the manuscript scientifically sound in its present form?

No

Are the interpretations and conclusions justified by the results?

No

Is the language acceptable?

Yes

Is it clear how to access all supporting data?

Yes

Do you have any ethical concerns with this paper?

No

Have you any concerns about statistical analyses in this paper?

Yes

Recommendation?

Major revision is needed (please make suggestions in comments)

Comments to the Author(s)

Studying in the real reservoir conditions is helpful to carry out the field work. I think that some major improvement can be made.

(1) In page 13, line 20. The first assumption is that the Berea cores used in these experiments are isotropic. But to the best of my knowledge, there are still some anisotropic Berea sandstones. So please give some pictures of the cores and setup.

(2) In page 13, line 30-31. Please explain the reason about which model is adapted in these experiments.

(3) In page 17, line 40. There are 23 groups of data in the table 1, but there are only 20 data points in figure 5. It should be pointed out that this is applicable to sandstone with a permeability of 100 mD.

(4) The conclusion in this manuscript is too simple. The effects of temperature and permeability on CO₂ diffusion coefficient are well-known. I think the authors need to facet something new with comparison to previous work done.

Decision letter (RSOS-181902.R0)

11-Mar-2019

Dear Mr Lv,

The editors assigned to your paper ("Experimental study of supercritical CO₂ diffusion coefficient in porous media under reservoir conditions") have now received comments from reviewers. We would like you to revise your paper in accordance with the referee and Associate Editor suggestions which can be found below (not including confidential reports to the Editor). Please note this decision does not guarantee eventual acceptance.

Please submit a copy of your revised paper before 03-Apr-2019. Please note that the revision deadline will expire at 00.00am on this date. If we do not hear from you within this time then it will be assumed that the paper has been withdrawn. In exceptional circumstances, extensions may be possible if agreed with the Editorial Office in advance. We do not allow multiple rounds of revision so we urge you to make every effort to fully address all of the comments at this stage. If deemed necessary by the Editors, your manuscript will be sent back to one or more of the original reviewers for assessment. If the original reviewers are not available, we may invite new reviewers.

- Data accessibility

If you wish to submit your supporting data or code to Dryad (<http://datadryad.org/>), or modify your current submission to dryad, please use the following link:
<http://datadryad.org/submit?journalID=RSOS&manu=RSOS-181902>

- **Competing interests**

- **Authors' contributions**

- **Acknowledgements**

- **Funding statement**

Kind regards,

Royal Society Open Science Editorial Office
Royal Society Open Science
openscience@royalsociety.org

on behalf of Professor R. Kerry Rowe (Subject Editor)
openscience@royalsociety.org

Associate Editor's comments:

The reviewers have a number of comments and suggestions for improvement, including revising the quality of the written language (please seek advice from services such as <https://royalsociety.org/journals/authors/language-polishing/>). You must also tackle the scientific concerns raised.

If you choose to submit a revised manuscript, the paper will be sent back to the original referees for further assessment -- if the revised manuscript does not satisfy the referees that the paper is

ready for publication, the paper may be rejected, as the journal does not routinely allow multiple rounds of major revision to be performed.

Comments to Author:

Reviewers' Comments to Author:

Reviewer: 1

Comments to the Author(s)

This paper presents experimental results measuring diffusion coefficients for supercritical CO₂ under a variety of pressure, temperature, and permeability conditions of Berea sandstone. These results add important data to other studies of supercritical CO₂ diffusion coefficients because the experiments were carried out under conditions relevant for CO₂ sequestration. The authors compile their results with very useful empirical relationship between the diffusion coefficient and pressure, temperature, and hydrocarbon viscosity. Such a laboratory-based relationship could prove very useful for numerical modeling studies of CO₂ sequestration, and therefore I recommend that these results be published.

The paper is written in a very straight-forward manner; "this is what we did, and this is what we measured." However, there are numerous grammatical and style problems, which could be fixed in revision. I list a few of them here:

line 42 (abstract): "...then grow...". Change to " then grows significantly with increasing pressure":

page 10.

line 4: change to "has recently attracted much attention".

line 23: Change "researches" to "studies". (apply this throughout the paper).

line 31: need citation after "Renner"

line 35: change "were 5.86...." to "were determined for conditions of 5.86..."

page 11.

line 14. Do not start sentence with And

page 13.

line 47: change to "is shown from other studies...".

page 14.

line 8: Change to "Figure 2 shows that the original...."

line 9: Change diffuse to diffusion

line 40: Change to "Figure 3a shows...."

page 15.

line 57. Feed is not capitalized (and not even sure if feed is the right word).

page 17.

line 43: change to "Figure 6 shows..."

line 57: change "literatures" to "studies"

page 19.

line 43: "different mixtures systems.."

line 44: change "were" to "are"

line 47-48. This sentence makes no sense

line 54: Moutos needs a citation

page 20.

line 5: change "had" to "have"

Reviewer: 2

Comments to the Author(s)

Studying in the real reservoir conditions is helpful to carry out the field work. I think that some major improvement can be made.

(1) In page 13, line 20. The first assumption is that the Berea cores used in these experiments are isotropic. But to the best of my knowledge, there are still some anisotropic Berea sandstones. So please give some pictures of the cores and setup.

(2) In page 13, line 30-31. Please explain the reason about which model is adapted in these experiments.

(3) In page 17, line 40. There are 23 groups of data in the table 1, but there are only 20 data points in figure 5. It should be pointed out that this is applicable to sandstone with a permeability of 100 mD.

(4) The conclusion in this manuscript is too simple. The effects of temperature and permeability on CO₂ diffusion coefficient are well-known. I think the authors need to facet something new with comparison to previous work done.

Author's Response to Decision Letter for (RSOS-181902.R0)

See Appendix A.

RSOS-181902.R1 (Revision)

Review form: Reviewer 2

Is the manuscript scientifically sound in its present form?

Yes

Are the interpretations and conclusions justified by the results?

Yes

Is the language acceptable?

Yes

Is it clear how to access all supporting data?

Yes

Do you have any ethical concerns with this paper?

No

Have you any concerns about statistical analyses in this paper?

No

Recommendation?

Accept as is

Comments to the Author(s)

Well revised.

Decision letter (RSOS-181902.R1)

10-May-2019

Dear Mr Lv,

I am pleased to inform you that your manuscript entitled "Experimental study of the supercritical CO₂ diffusion coefficient in porous media under reservoir conditions" is now accepted for publication in Royal Society Open Science.

Kind regards,

Andrew Dunn

on behalf of Prof R. Kerry Rowe (Subject Editor)

Associate Editor Comments to Author:

Thank you for the revised paper -- the reviewers are now satisfied the paper may be accepted.

Reviewer comments to Author:

Reviewer: 2

Comments to the Author(s)

Well revised.

Appendix A

Dear Editor and Reviewers

Thank you for your comments concerning our manuscript entitled “**Experimental study of supercritical CO₂ diffusion coefficient in porous media under reservoir conditions**”. The comments are all valuable and helpful for improving our paper and our research. We provide this response letter to explain, point by point, the details of our revisions in the manuscript and our responses to the reviewers’ comments as follows. In order to make the changes easily traceable for you and reviewers, all the revisions were in red font both in this letter and the word file of manuscript we submitted. We also used language polishing service provided by *American Journal Expert* recommended by editor to improve the written English. The certification verification key is 0BD9-7416-C394-725D-5FEC.

We hope the revised manuscript would meet your and reviewers’ requirements.

Sincerely

Junchen Lv

School of Energy and Power Engineering

Dalian University of Technology

Reviewer 1

line 42 (abstract): "...then grow...". Change to " then grows significantly with increasing pressure"

Responses to comments: Thanks for the reviewer's suggestion. We followed the reviewer's comments and make corresponding changes. The sentence has been rephrased to "The supercritical CO₂ diffusion coefficient first increases slowly at 10 MPa and then **grows significantly with increasing pressure.**"

page 10.

line 4: change to "has recently attracted much attention".

line 23: Change "researches" to "studies". (apply this throughout the paper).

line 31: need citation after "Renner"

line 35: change "were 5.86..." to "were determined for conditions of 5.86..."

Responses to comments: Thanks for the reviewer's suggestion. We followed the reviewer's comments and make corresponding changes.

1. The sentence has been rephrased to "Global warming caused by the excessive emission of greenhouse gases (mainly CO₂) has **recently attracted much attention.**"
2. We change all "researches" to "studies" throughout the paper.
3. As for the citation after "Renner", we put the reference right after that sentence to make sure the reference format keeps same throughout the paper.
4. The sentence has been rephrased to "The experimental diffusion coefficients of CO₂ and ethane in decane **were determined for conditions** of 5.86 MPa and 4.14 MPa at 311.15 K, respectively."

page 11.

line 14. Do not start sentence with And

Responses to comments: Thanks for the reviewer's suggestion. We followed the reviewer's comments and replace "and" with "moreover". The sentence has been rephrased to "**Moreover,** a pressure-temperature-viscosity based empirical equation for the supercritical CO₂ diffusion coefficient in hydrocarbon saturated porous media was developed."

page 13.

line 47: change to "is shown from other studies..."

Responses to comments: Thanks for the reviewer's suggestion. We followed the reviewer's comments and make corresponding changes. The sentence has been rephrased to "A simplified expression of **the** CO₂ diffusion coefficient under non-swelling conditions is shown in Eq. (2), which **is obtained from other studies**"

page 14.

line 8: Change to "Figure 2 shows that the original..."

line 9: Change diffuse to diffusion

line 40: Change to "Figure 3a shows...."

Responses to comments: Thanks for the reviewer's suggestion. We followed the reviewer's comments and make corresponding changes.

1. The sentence has been changed to "Figure 2 shows that the original experiments were in good agreement with repetitive experiments."
2. The sentence has been changed to "The pressure data recorded during the supercritical CO₂ diffusion process for Experiment No. 1 are shown in Figure 3."
3. The sentence has been changed to "Figure 3(a) shows that the pressure-time (P-T) profile can be separated into three regions"

page 15.

line 57. Feed is not capitalized (and not even sure if feed is the right word).

Responses to comments: Thanks for the reviewer's suggestion. We replace "feed" with "experiment" throughout the paper.

page 17.

line 43: change to "Figure 6 shows..."

line 57: change "literatures" to "studies"

Responses to comments: Thanks for the reviewer's suggestion. We followed the reviewer's comments and make corresponding changes.

1. The sentence has been changed to "Figure 6 shows that the diffusion coefficient of the supercritical CO₂ increases with pressure."
2. The sentence has been changed to "Similar experimental observation could be found in other studies."

page 19.

line 43: "different mixtures systems.."

line 44: change "were" to "are"

line 47-48. This sentence makes no sense

line 54: Moutos needs a citation

Responses to comments: Thanks for the reviewer's suggestion. We followed the reviewer's comments and make corresponding changes.

1. The sentence has been changed to "The experimental results of the CO₂ diffusion coefficient in different mixture systems"
2. The sentence has been changed to "experimental conditions are listed in Table 2"
3. We rephrased the sentence and changed it to "The differences in results were caused by the state of CO₂ in the different experiments."

4. As for the citation after “Moutos”, we put the reference right after that sentence to make sure the reference format keeps same throughout the paper.

page 20.

line 5: change "had" to "have"

Responses to comments: Thanks for the reviewer’s suggestion. We followed the reviewer’s comments and make corresponding changes. The sentence has been changed to “The experimental results in this paper **have** better practical meaning since the pressure and temperature conditions simulated the underground reservoir conditions.”

Reviewer 2

In page 13, line 20. The first assumption is that the Berea cores used in these experiments are isotropic. But to the best of my knowledge, there are still some anisotropic Berea sandstones. So please give some pictures of the cores and setup.

Responses to comments: Thanks for the reviewer's suggestion. We took some photos of the Berea cores and the experimental setup, as shown in Figure 1. The merchant who provided Berea cores confirmed that these cores were made isotropic. Those cores marked with 50 are cores with permeability of 50 mD. The other three marked with 100 are cores with permeability of 100 mD.

Figure 1. Berea cores we used in the experiment.

Figure 2. experiment setup.

In the left of Figure 2, it's the experiment system we build to measure the supercritical CO₂ diffusion coefficients. From left to right, there are computer which collects the pressure data in real time, temperature and pressure processor box, Julabo oil bath and

pump. In the right of Figure 2, it's the customized diffusion cell made of stainless steel and can sustain pressure up to 30 MPa and temperature up to 150 °C. On the side of the cell, there is an inlet which connected to the pressure and temperature transducer. The second inlet on the top is the one we used to inject and eject the supercritical CO₂.

In page 13, line 30-31. Please explain the reason about which model is adapted in these experiments.

Responses to comments: Thanks for the reviewer's suggestion. The model we used in this study is adopted from Li's work which we referred in the manuscript. The reason that we used this model is that the cores were sealed on the top and bottom sides in both his and our study. So the diffusion process became a natural convection mass transfer induced by the diffusion CO₂ into a decane-saturated vertical porous column. Equation 2 is a simplified version after a series of derivation from equation 1 and it is really convenient for calculation in practical application.

We followed the reviewer's comments and make corresponding changes.

The paragraph has been changed to “**Based on the treatments to the cores mentioned previously, the diffusion process in this study became a natural convection mass transfer process induced by the diffusion of CO₂ into a liquid-saturated vertical porous column.** The effective CO₂ diffusion coefficient under non-swelling conditions can be obtained from Fick's first law and the continuity equation, as shown in Eq. (1).”

Reference:

Li, S.; Li, Z.; Dong, Q., Diffusion coefficients of supercritical CO₂ in oil-saturated cores under low permeability reservoir conditions. *Journal of CO₂ Utilization* **2016**, 14, 47-60.

In page 17, line 40. There are 23 groups of data in the table 1, but there are only 20 data points in figure 5. It should be pointed out that this is applicable to sandstone with a permeability of 100 mD.

Responses to comments: Thanks for the reviewer's suggestion. The correlation proposed is based on the data acquired with Berea cores with permeability of 100 mD. That's the reason why there are only 20 data points in figure 5. We followed the reviewer's comments and make corresponding changes. The sentence has been rephrased to “The correlation for Berea cores **with a permeability of 100 mD** is shown as Equation 3.”

The conclusion in this manuscript is too simple. The effects of temperature and permeability on CO₂ diffusion coefficient are well-known. I think the authors need to facet something new with comparison to previous work done

Responses to comments: Thanks for the reviewer's suggestion. The major novelty of this study focus on supercritical CO₂ diffusion characteristics in oil-saturated porous

media under reservoir conditions. There are few studies of CO₂ diffusion process under high pressure and temperatures with the presence of porous media. Also, we are aiming at finding out whether the effects of temperature, pressure and permeability would still be the same when it comes to reservoir condition. We rephrased the conclusion part to emphasize the novelty of this study and add more comparison with previous work regarding with the presence of porous media.

The paragraph has been rephrased to “These results demonstrated that **under reservoir conditions** the supercritical CO₂ diffusion coefficient **in oil-saturated porous media** increased with increasing pressure and temperature, respectively. **However, the effect of pressure on the diffusion process decreased at elevated temperature condition. Also, the supercritical CO₂ diffusion coefficient first increases slowly at 10 MPa and then grows significantly with increasing pressure.** The increase in the core permeability increased the diffusion coefficient and the growth trend was steady throughout the whole temperature range in the experiments. **The supercritical state of the CO₂ and the presence of porous media significantly impeded the diffusion process, compared with the results from pure gas CO₂ diffusion in bulk alkanes in previous study.** The pressure-temperature-viscosity based empirical equation developed in this paper successfully predicted the supercritical CO₂ diffusion coefficients in porous media under reservoir conditions. Moreover, all the experimental data contributed to the theoretical study of supercritical CO₂ diffusion in porous media at elevated **temperatures and pressures.**”